# Detect Overlapping Community Based on the Combination of Local Expansion and Label Propagation

**Xu Li** [1,*] and **Qiming Sun** [2]

1 Applied Economics, Beijing University of Posts and Telecommunications, Beijing 100876, China
2 Management Science and Engineering, Beijing University of Posts and Telecommunications, Beijing 100876, China; sunqm@bupt.edu.cn
* Correspondence: chunxiaol@bupt.edu.cn; Tel.: +86-138-1140-1422

**Abstract:** It is a common phenomenon in real life that individuals have diverse member relationships in different social clusters, which is called overlap in the science of network. Detecting overlapping components of the community structure in a network has extensive value in real-life applications. The mainstream algorithms for community detection generally focus on optimization of a global or local static metric. These algorithms are often not good when the community characteristics are diverse. In addition, there is a lot of randomness in the process of the algorithm. We proposed a algorithm combining local expansion and label propagation. In the stage of local expansion, the seed is determined by the node pair with the largest closeness, and the rule of expansion also depends on closeness. Local expansion is just to obtain the center of expected communities instead of final communities, and these immature communities leave only dense regions after pruning according to certain rules. Taking the dense regions as the source makes the label propagation reach stability rapidly in the early propagation so that the final communities are detected more accurately. The experiments in synthetic and real-world networks proved that our algorithm is more effective not only on the whole, but also at the level of the node. In addition, it is stable in the face of different network structures and can maintain high accuracy.

**Keywords:** complex networks; overlapping community; local expansion; label propagation

## 1. Introduction

In real-world social networks, the individual behavior is not simply obedient to the collective. Restricted by diverse social relations, they often naturally gather into some social structure, named a community in network research [1]. The community structure is considered to be the functional unit in a network system, as it often accounts for the functionality. Although there is no unified definition of community, many technologies have been developed to detect community, such as modularity maximization [2], random walks [3], particle swarm optimization [4], spectral clustering [5], etc. In the early studies of community detection, much of the focus was on disjoint communities, where individuals have only one membership [6]. However, it is obvious that people's social memberships are not singular. For example, a person could be a member of family while also belonging to other social groups, such as those of friends and colleagues. Moreover, in large-scale social networks, the membership of an individual is more complex.

In general, the characteristic of nodes being able to belong to multiple communities is called overlap. There are extensive works on detecting overlapping communities [7] for which we provide a more detailed introduction in the next section. However, these studies are still insufficient in some ways. First, the static local optimization goal for detecting overlapping communities is adopted generally, such as fitness function in local expansion [8], optimization for modularity [9] and local Nash equation [10] in the game algorithm. The density, scale and edge distribution of different communities in real social networks are diverse. A static optimization goal can just be applied to detect some communities, and

not to the whole. Second, there is a lot of randomness in the detection. For example, the integration order of a clique has no rules in most algorithms based on clique percolation. The selection of the start position in local expansion is always random. Some updated rules of labels in label propagation are not strict when nodes face the same degree of label influence, which may lead to unstable results. The existence of randomness often means that the algorithm performance can be optimized by some beneficial rules.

In this paper, an algorithm that combines local expansion and label propagation (*LELP*) is proposed. We use an extended local expansion to obtain the immature communities instead of the final communities so that the disadvantages of static optimization are avoided. The seed of local expansion is selected based on the node pair with the highest *closeness*, an index expanded from the Jaccard coefficient [11], which governs the direction of expansion. The immature communities detected by local expansion are pruned to be dense regions in order to conduct label propagation, a speaker–listener-based information propagation process. As the source of propagation, these dense regions can reduce the process of reaching local stability in the early stage of label propagation, making this dynamic process more accurate. We conduct experiments on both synthetic and real-world networks to compare our algorithm with others. The experimental results show that our algorithm generally performs well and is stable in different conditions.

## 2. Related Works

In this section, several detection algorithms for overlapping community are reviewed and classified based on characteristics that reflect how communities are detected.

The idea of local expansion is to generate natural communities based on the local structure. This kind of algorithm constantly expands the seed by optimizing the fitness function. The form of fitness function and the rules for selecting the seed have a great influence on the performance; the two are diverse in different algorithms. Iterative scan (*IS*), a two-step local expansion process, was proposed by Baumes et al. [8], where nodes are ranked according to the page rank [12] first and then removed in order of ranking. The part of the nodes that are far away from each other serve as seeds. The fitness function adopted is the following:

$$f(c) = \frac{w_{in}^c}{w_{in}^c + w_{out}^c},$$ (1)

where $w_{in}^c$ and $w_{out}^c$ are the total weight of the internal and external edges of the community, respectively. After each expansion, nodes in the current cluster are removed if the fitness function can be optimized. This action often divides a community into several disconnected parts, which could lead to interruption of the expansion before optimization. So, Kelly proposed an improved algorithm—connected iterative scan (*CIS*) [13]—where only the one that maximizes the fitness function is reminded after the community is split. In addition, *CIS* extends the fitness function with $\lambda e_p$, where $\lambda$ controls how the algorithm behaves in areas with different sparsity of edges, and $e_p$ is the edge probability. The resolution problem of local expansion was pointed out by maximization of local fitness (*LFM*) [14], which argues that the size and internal density of a community are diverse. It is unreasonable to adopt the same fitness function to control expansion on different networks. Therefore, it proposed a fitness function that can adjust the resolution, as follows:

$$f(c) = \frac{k_{in}^c}{(k_{in}^c + k_{out}^c)^\alpha},$$ (2)

where $k_{in}^c$ and $k_{out}^c$ are the total internal and external degree of the community. Parameter $\alpha$ controls the algorithm performance in different sparsities, but the setting rules of $\alpha$ cannot be interpreted. It determines the range of $\alpha$ by a grid search with high computation costs. In addition, *LFM* takes random nodes as initial seeds. It is obvious that a single node must expand successfully at the first attempt. For this reason, the merging of overlapping natural communities (*MONC*) adds a constant 1 to the denominator of Equation (2),

which makes the algorithm more stable [15]. Neighborhood-inflated seed expansion (*NISE*) develops new seeding strategies based on Page Rank that optimizes the community score [16]. Its main idea is embodied in the step of neighborhood inflation, where seeds can represent the characteristics of their entire neighborhood. In general, local expansion algorithms usually expand from the dense area of edge or degree to the sparse area so that the seed always plays an important role. The number, quality and location of the seed will affect the partitions significantly. If we can determine the seed in the center of the expected community at the beginning, the detection quality could be naturally higher. The optimization objective of local expansion is static, where the same fitness function is applied from beginning to end, so the quality of the communities detected are often poor, due to the diversity in the size and structure in the networks.

Label propagation is a dynamic algorithm based on the speaker–listener information spread process. Nodes can propagate and accept labels with information of the local structure. Through the interaction of labels, nodes with the same structure are classified into one community. Label propagation algorithm (*LPA*), proposed by Barber et al., is a classical algorithm based on this idea [17]. Initially, each node is given a unique label. In each propagation, the label of a node $i$ is replaced by the most common label in neighborhood $c$, as shown in the following:

$$l(i) = l(\max_{j \in c}\{l(j)\}). \tag{3}$$

The labels are updated constantly until they reach a stable state, and nodes with the same label form a community. The community overlap propagation algorithm (*COPRA*) allows nodes to have multiple labels [18]. The membership of a node is no longer hard but rather a series of labels with different attribution coefficients. However, the maximum number of communities that a node can belong to is limited. In the speed label propagation algorithm (*SLPA*), proposed by Xie, nodes update their labels based on pairwise interaction rules [19]. In each propagation, the node no longer passively receives the neighbors' labels to update its own, but interacts with the neighbors and shares information. In addition, the history labels of each iteration are recorded and nodes update labels based on the previous information. The label propagation algorithm can adapt to the diverse community structure, but the general issue is that local label always reaches stability quickly in the simple structure. Given that stable nodes in the early stages have a significant impact on the subsequent propagation, the performance of algorithms will be poor if propagation starts from an inappropriate place, such as the border of the expected community and several dense substructures in the expected community. Contrarily, if the labels of nodes in the center area of the expected community reach stability first, the communities could be detected more accurately.

The clique percolation method (*CPM*) argues that a community is composed of many small cliques, and there are common members among adjacent cliques [20]. Through these shared members, small cliques are connected together to form a community. *CFinder* proposed by Palla is an implementation of *CPM* [1]. First, it searches for maximal cliques of size $K$. If two adjacent k-cliques share k-1 members, they will merge into a new component. In the process of percolation, more cliques connect together and nodes shared by multiple cliques are regarded as overlapping. Agglomerative hierarchical clustering based on maximal clique (*EAGLE*) takes maximal k-cliques as the seeds and simply employs the aggregate framework to detect the community [21]. Specially, there is a step called community merging for aggregating two cliques with plenty of identical nodes in case too many nodes are treated as overlapping. However, this kind of algorithm depends on parameter $K$. Too- large $K$ could lead to a lot of trivial, isolated cliques, which hinders the detection of large communities. Otherwise, it also fails to terminate in many large social networks when the $K$ is too small.

There are other interesting algorithms. For example, a game-theoretic framework is proposed in Chen et al. [22]. In this theory, each node is regarded as an agent who can choose to join, quit and change his/her community affiliation. The goal of each agent is to

maximize his/her own utility, which is determined by the gain function and loss function. A community is associated with a Nash local equity. When the local equilibrium is reached, the members of the community are determined. Zhou et al. proposed an extended method of Chen's work [23]. They take two strategies instead of the action set for obtaining a new label of each agent, and a gain function with similarity is adopted. The difference between the detection algorithms based on game theory mainly lies in the utility functions. In order to prevent confusion, the algorithm named *Game* mentioned in this paper refers to that of Zhou et al. Another kind of algorithm is based on generative stochastic modeling, such as internal and external association to detect communities (*IEDC*), whose general framework is based on the principle of dividing each community into non-overlapping and overlapping parts [24]. The probabilistic membership of a primary node consists of the internal association degree along with the external association degree, which, together, control the membership of nodes.

Different from the above algorithms, our algorithm combines the advantages of dynamic and static algorithms. We extend the local expansion with a new fitness function based on a new metric named *closeness* to obtain a dense region rather than the final community in [13–15]. Although the community detected by local expansion in the face of diverse structures is not good due to static optimization objectives, the local expansion that is provided with an appropriate seed by *closeness* can expand from dense to sparse region, orderly. To ensure that the density region is almost the same as the center of the expected community, a process called prune is adopted to remove some unimportant nodes from the immature community generated by local expansion, which provides a more accurate starting point for the label than the models in [18,19]. Moreover, label propagation that takes the density region as the source is conducted, where the labeled set of nodes is updated by those labels whose frequency is greater than the average to reduce randomness.

## 3. Algorithm

The essence of our algorithm is the combination of local expansion and label propagation (*LELP*). It first intends to detect immature communities based on the improved local expansion. These immature communities are pruned to obtain dense regions that are likely to be the central area of the expected communities. Then, the label propagation that takes the dense regions as a source is conducted, where nodes are allowed to carry multiple labels. After propagation, nodes with the same label form a community and nodes with multiple labels are identified as overlapping.

In the stage of local expansion, a new fitness function, different from the previous ones that are related to only edges, is adopted to evaluate expansion. The basic assumption behind it is that a community essentially comprises local structures, involving those close nodes. It is difficult to determine the relationship between two nodes depending only on edges. Therefore, we introduce a new metric named *closeness* to measure the relationship of nodes. The *closeness* is defined as the number of neighbors shared by two nodes divided by the number of neighbors in their union, shown as follows:

$$closeness_{(d)}(i,j) = \frac{cover(i,d) \cap cover(j,d)}{cover(i,d) \cup cover(j,d)}, \tag{4}$$

where $cover(i,d)$ is the neighborhood of node $i$ and parameter $d$ is a positive integer governing the range of neighbors. The *closeness* is the same as the Jaccard coefficient when $d$ is equal to 1 [25]. The larger the $d$, the wider the range of neighbors considered. However, the relationship of the population will decline with distance in the real world, so $d$ should be set to a small integer. In addition, two nodes that are not directly connected to each other may have common second-order neighbors, which means that their second-order closeness is not 0. For this case, the *closeness* between two nodes without direct edges is uniformly set to 0. In this way, we can weight edges by the *closeness* according to Equation (4) and generate a new weighted network.

Local expansion is no longer dependent on random seeds in *LELP*. Random seeds mean that the starting position of the local expansion is random, which leads to unstable results in repeated experiments. There is no guarantee that the local expansion could start from an appropriate place, so the quality of the partition is often poor. In general, the node pairs with a high ratio of common neighbors tend to be located in the center of the expected community, while those with a low ratio of common neighbors tend to be located in the border. For this reason, node pairs are sorted according to *closeness*, and those with the largest *closeness* are determined as seeds first in order to ensure an optimal starting point of the local expansion. Therefore, the fitness function is modified as follows:

$$f(c) = \frac{closeness_{in}^c}{closeness_{in}^c + closeness_{out}^c},$$

(5)

where $closeness_{in}^c$ and $closeness_{out}^c$ are the total internal and external *closeness* of the cluster. When a round of local expansion ends, the next round starts from another seed that has not yet expanded.

In the previous works of detecting an overlapping community based on local expansion methods, such as *CIS, LFM, MONC*, the detection is over after the above process. These algorithms always have low quality because the fitness function is static during the entirety of the detection. In fact, the effect of detecting is not always good when the same fitness function is used to detect communities with different sizes. A diverse scale of communities in networks affects the algorithm accuracy, especially at the border of communities. Unlike previous algorithms, the purpose of local expansion adopted in *LELP* is not to generate the final communities, but to obtain the central area of expected communities, which are named the dense region. These dense regions are used as the source for label propagation in the following.

Then, we adopt an operation called prune to remove the unimportant component of immature community detected by local expansion. All nodes in one immature community are scanned, and the nodes for which more than half its neighbors are not inside are removed first. Then, those nodes that can increase the fitness function if they are removed are removed as well. In this way, the immature communities could be pruned to the dense regions, of which the number is close to that of the final communities. Nodes inside the dense regions are regarded as internal nodes that are closely related to each other, and nodes outside the dense regions are named boundary nodes.

Finally, label propagation is conducted based on the dense regions. Initially, the internal nodes in each dense region are labeled with the same label and set to a passive state, while each external node is labeled with a unique label and is set to an active state. In the process of label propagation, each active node updates its own label, according to the frequency of its neighbors' labels. A node is allowed to have multiple labels when there are several labels whose frequency is larger than or equal to the average. If the label of a node does not change after a propagation, the node becomes passive, and it no longer accepts the labels of its neighbors. This process is repeated until all nodes in the network become passive. When label propagation stops, the labels of the nodes indicate the memberships. Nodes with the same label form a community, and nodes with multiple labels overlap.

Because the dense regions obtained from the local expansion are taken as the source, label propagation could spread from the center of the expected communities to the border in an orderly way. It could avoid the labels of nodes in the border of the communities with a simple structure reaching stability first, which would lead to the failure of the label propagation, locally. At the same time, the labels of nodes inside dense regions are already stable before label propagation, so it does not cost a lot of time to reach local stability. In short, the combination of local expansion and label propagation can complement each other.

In this way, the process of *LELP* can be interpreted as the following procedure:

Step 1. Local expansion based on *closeness*. The *closeness* between each pair of directly connected nodes are measured by Equation (4). If there is no direct edge between two nodes, the *closeness* is set to 0. Then, node pairs are sorted according to *closeness* and local

expansion is conducted from the pair with the largest *closeness*. An immature community is generated when a round of expansion is optimal, and the next pair in order of *closeness* that has not been expanded is selected as the seed. Local expansion is iterated until all nodes in the network are tried. The two immature community with more than half the same components are merged into one.

Step 2. The immature community is pruned to the dense region. Each node in the immature community generated by Step 1 is scanned. Those nodes for which more than half of its neighbors are not in the same immature community are removed. In addition, nodes are also removed if the fitness can be improved. This process is called pruning. The immature communities that are pruned are defined as dense regions. Nodes inside dense regions are called internal nodes and the rest are called boundary nodes.

Step 3. Label propagation. The internal nodes in a dense region are labeled with the same label and set to be passive, while the external nodes are labeled with unique labels and set to be active. In the process of label propagation, each active node updates the label by the frequency of its neighbors' labels. A node is allowed to have multiple labels. Label propagation is repeated until all nodes reach a stable state. The nodes with the same label are grouped into a community, and those with multiple labels are identified as overlapping.

## 4. Experimental Result

### 4.1. Evaluation Criteria

The normalized mutual information ($NMI$) is used to evaluate algorithms by comparing the similarity between the ground truth and detected community [26]. The extended $NMI$ to evaluate the algorithm for overlapping communities was proposed by Lancichinetti [14]. The membership of the node $i$ that belongs to multiple communities can be regarded as a binary array with $|C'|$ entries, which is the community number of partition $C'$. The $k$-th entry can be represented as a random variable $X_k = (\mathbf{X})_k$. If node $i$ belongs to the $k$-th community $C'_k$, $(x_i)k = 1$; otherwise, $(x_i)k = 0$. Then, the probability distribution can be obtained by the number of nodes in $C'_k$ and all nodes. The same holds for the random variable $Y_l$ associated with the community $C''_l$ of partition $C''$. Due to the independence of $X_k$ and $Y_l$, the joint distribution and conditional distribution can be obtained. We can define the amount of information to infer $X_k$ given a certain $Y_l$ as $H(X_k|Y_l) = H(X_k, Y_l) - H(Y_l)$, and its minimum can be defined as the conditional entropy of $X_k$ with respect to all the components of $\mathbf{Y}$ for which the average is $H(\mathbf{X}|\mathbf{Y})_{norm}$. Finally, the extended $NMI$ is defined as follows:

$$N(\mathbf{X}|\mathbf{Y}) = 1 - \frac{1}{2}[H(\mathbf{X}|\mathbf{Y})_{norm} + H(\mathbf{X}|\mathbf{Y})_{norm}], \qquad (6)$$

which is between 0 and 1. The function $N(\mathbf{X}|\mathbf{Y})$ is equal to 1 if and only if $X_k = f(Y_l)$ for a certain $l$, and vice versa.

The extended $NMI$ can only evaluate the algorithm of detecting overlapping communities at the overall level, but it cannot work at the node level. Therefore, we adopt the F-score to measure the accuracy of the algorithms at the node level [27]. The F-score is defined as the harmonic mean of *precision* and *recall*, as follows:

$$\text{F-score} = \frac{2 \cdot precision \cdot recall}{precision + recall}, \qquad (7)$$

where *precision* represents the proportion of the detected overlapping nodes that are true overlapping nodes, and *recall* is defined as the number of correctly detected overlapping nodes divided by the true number of overlapping nodes. The F-score accounts for the balance between the detection quantity and quality, and it reaches its best and worst values at 1 and 0, respectively.

The performance of algorithms can be measured by $NMI$ and the F-score when the ground truth of the community is known. Given that the ground truth of most real-

world networks are not available, we adopted an extended modularity: the overlapping modularity $Q_{ov}^e$ proposed in [28], as our measure. $Q_{ov}^e$ takes the membership of a node as a weight for modularity. The function is as follows:

$$Q_{ov}^e = \frac{1}{2m} \sum_c \sum_{i,j \in c} [A_{ij} - \frac{k_i k_j}{2m}] \frac{1}{O_i O_j}, \tag{8}$$

where $O_i$ is the number of memberships that node $i$ owns. One may argue that $Q_{ov}^e$ is close to extended modularity $Q_{ov}^{ne}$, proposed by Nepusz et al. [29], where the degree of membership of node $i$ in the community is the substitution of $O_i$ in $Q_{ov}^e$.

*4.2. Synthetic Networks*

It is necessary to have benchmarks to observe the performance of our algorithm and compare it with others. Due to the lack of ground truth, real-world networks do not show clear contrast results but the synthetic networks are adaptable. We adopt *LFR* benchmarks proposed in [30] and set the parameters of the *LFR* generator referring to [14]: the size of networks are between 1000 and 5000. The average degree is set to 10, which is the same as most real-world social networks. The node degree and community size are governed by power law distributions with exponents $\tau_1 = 2$ and $\tau_2 = 1$, respectively. The maximum degree is $k_{max} = 50$, and community sizes vary within a small range $s = (10, 50)$ or large range $l = (20, 100)$. The mixing parameter $\mu$ is set to 0.1 or 0.3. $O_n$ and $O_m$ together control the overlap, where the former is the number of overlapping nodes, while the latter is the number of communities to which each overlapping node belongs. $O_n$ is set to 10% or 50% of the total nodes so as to indicate the low or high density of overlapping nodes, respectively. $O_m$ varies from 2 to 8, indicating the diverse membership of overlapping nodes. By changing the combination of $O_n$ and $O_m$, we create detection tasks with different difficulties so that we can compare the performance of the algorithms in greater detail. The specific parameter settings of *LFR* benchmarks are shown in Table 1. For each set of parameters, we generate 100 instantiations and adopted the average performance of each algorithm. For each set of parameters, we generated 100 instantiations and 100 times experiments were carried out on each instantiation. We adopted the average of all these experiments as the algorithm performance.

**Table 1.** The detailed parameters of *LFR* benchmarks are shown. *LFR* 1 to *LFR* 4 are grouped to test the performance of algorithms under different network sizes and mixing parameters, while *LFR* 4 to *LFR* 7 are governed by different community sizes and overlapping ratios.

| Name | $N$ | $\bar{k}$ | $max_k$ | $\mu$ | $min_c$ | $max_c$ | $O_n$ | $O_m$ |
|---|---|---|---|---|---|---|---|---|
| LFR 1 | 1000 | 10 | 50 | 0.1 | 20 | 100 | 100 | 2–8 |
| LFR 2 | 1000 | 10 | 50 | 0.3 | 20 | 100 | 100 | 2–8 |
| LFR 3 | 5000 | 10 | 50 | 0.1 | 20 | 100 | 500 | 2–8 |
| LFR 4 | 5000 | 10 | 50 | 0.3 | 20 | 100 | 500 | 2–8 |
| LFR 5 | 5000 | 10 | 50 | 0.3 | 10 | 50 | 500 | 2–8 |
| LFR 6 | 5000 | 10 | 50 | 0.3 | 20 | 100 | 2500 | 2–8 |
| LFR 7 | 5000 | 10 | 50 | 0.3 | 10 | 50 | 2500 | 2–8 |

We compare *LELP* with some representative algorithms. For algorithms with tunable parameters, the setting that can obtain the best result is selected. For *CFinder*, $k$ varies from 3 to 8. For *LFM*, $\alpha$ ranges from 0.8 to 1.6 with an interval of 0.1. For *SLPA*, parameter $r$ varies from 0.05 to 0.5 with an interval of 0.05. For *LELP*, we measure its performance with different $d$, which controls the range of node neighbors in *closeness*. In general, the local structure information around the nodes is not fully reflected if $d$ is small, while the local details may be ignored if $d$ is too large. Therefore, we set $d$ from 1 to 4 to observe

the performance of *LELP* as shown in Figure 1. These parameters are still retained in real-world networks.

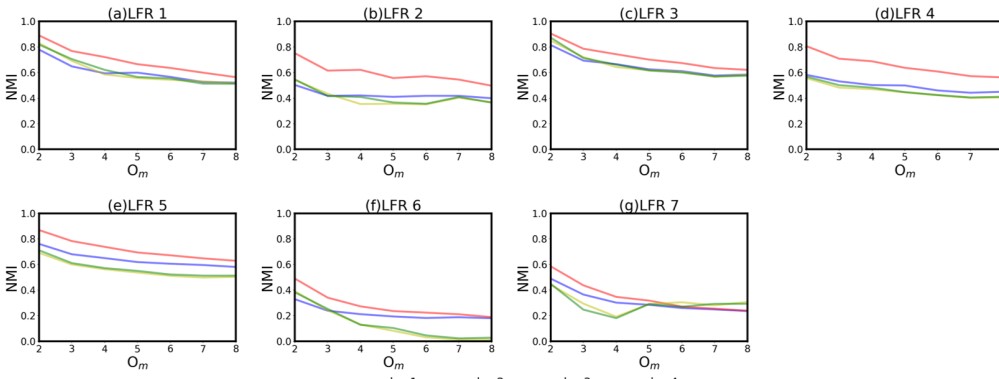

**Figure 1.** In the sub figure (**a**–**g**), the curves with different colors represent the NMI of LELP when *d* is different.

It can be seen that the performance of *LELP* with $d = 4$ is the worst in most cases and it is very close to that when $d = 3$. This may be due to the fact that the smaller local structure is ignored when the neighborhood is large. In addition, the ratio of shared neighbors tends to be stable as the scale of neighbors expands. The performance of *LELP* with $d = 1$ is slightly better than the former two cases in the face of small communities. In the cases where the proportion of the external edge is small ($\mu = 0.1$), there is little difference among *LELP* with different *d* (the four curves are very close in Figure 1a,c). In the networks where there are more external edges of nodes ($\mu = 0.3$), the *LELP* with $d = 2$ performs far better than others, especially when the community size is large and the membership is complex (the red curve is much higher than the others in Figure 1b,d, respectively). In general, the *LELP* with $d = 2$ performs better in most cases. Therefore, in the following of this paper, we choose the result of our algorithm with $d = 2$.

We first detect the performance of algorithms in the cases where there are different mixing parameter $\mu$ and network size $N$. The former controls the proportion of the internal edge to the external edge of nodes. Generally, the larger the $\mu$, the fuzzier the community structure. When $\mu$ is close to 0.5, the community structure is not prominent at all. Obviously, the experimental results are in line with this fact. When $\mu = 0.1$ but other conditions are the same, the *NMI* of all algorithms are almost better than those when $\mu = 0.3$ (the yellow and blue curves are at the top of most *LFR*s). For *LELP*, high $\mu$ does not weaken its performance greatly (the curves are very close in Figure 2a). Parameter $N$ determines the network size, that is, the total number of nodes in the network. The performance of algorithms in larger networks is slightly better than that of small networks, except *EAGLE*, which deteriorates greatly with large $N$. $O_m$ represents the membership of nodes. When $O_m$ is large, the task of the algorithm is difficult. As shown in Figure 2, the majority of the *NMI* curves of each algorithm gradually decreases with the increase in $O_m$.

Then, the *NMI* of the algorithms is measured in the cases where the number of overlapping nodes $O_n$ and community size are different. As expected, there is a serious decline in the performance of algorithms when there are a large number of overlapping nodes in networks (yellow and green curves are much lower than blue and red curves for most *LFR*s in Figure 3). This is because $O_n$ can indirectly determine the overlapping size among communities. A high overlapping size will hinder the accuracy of algorithms in detecting a community. Then, most algorithms perform better in the *LFR* with smaller communities. The reason may be that the structural characteristics of a community become blurred as the community size increases, which is also mentioned in [31,32]. This feature has a great influence on simple local expansion or label propagation. For the former, the sensitivity of the fitness function lowers as the community size increases. For the latter, there are so many dense structures within a large community that the labels can reach

stability at many places, which leads to the detection failure. Unlike others, *LELP* is less affected by the community size than ther other algorithms as shown in Figure 3a. This is because local expansion is only used in the first stage of *LELP* to locate dense regions, not the final community. These dense regions are small but have high density of edges, which prevents accuracy degradation. These dense regions determine the initial seed of label propagation in the following process, which can ensure that the label propagates from the dense to sparse regions. Usually, these two regions are likely to be the center and border of the expected community, respectively. Using the dense region instead of single node as the seed makes the labels reach stability more quickly in the early propagation.

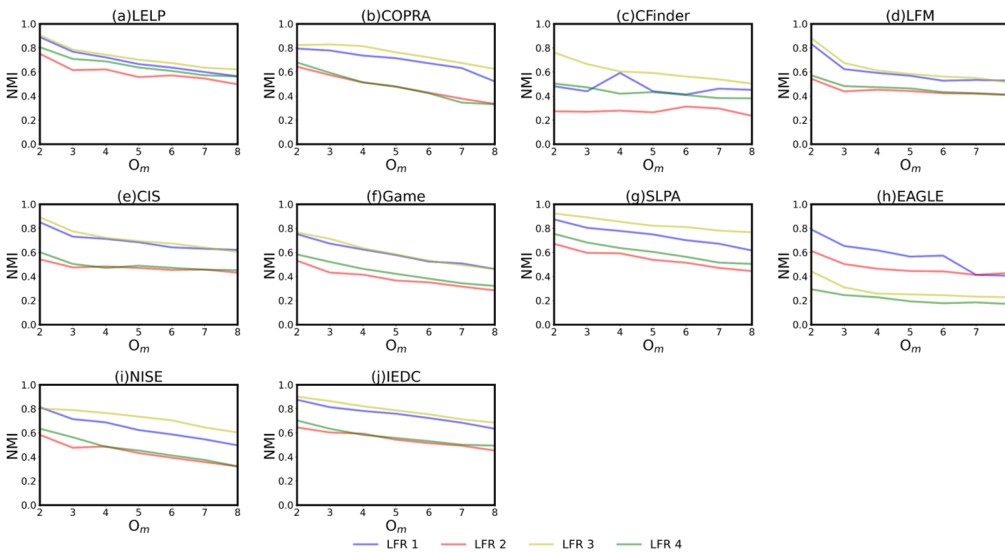

**Figure 2.** Sub figure (**a**–**j**) represent the NMI of different algorithms in the *LFR* 1 to *LFR* 4, where there are different mixing parameter $\mu$ and network size $N$.

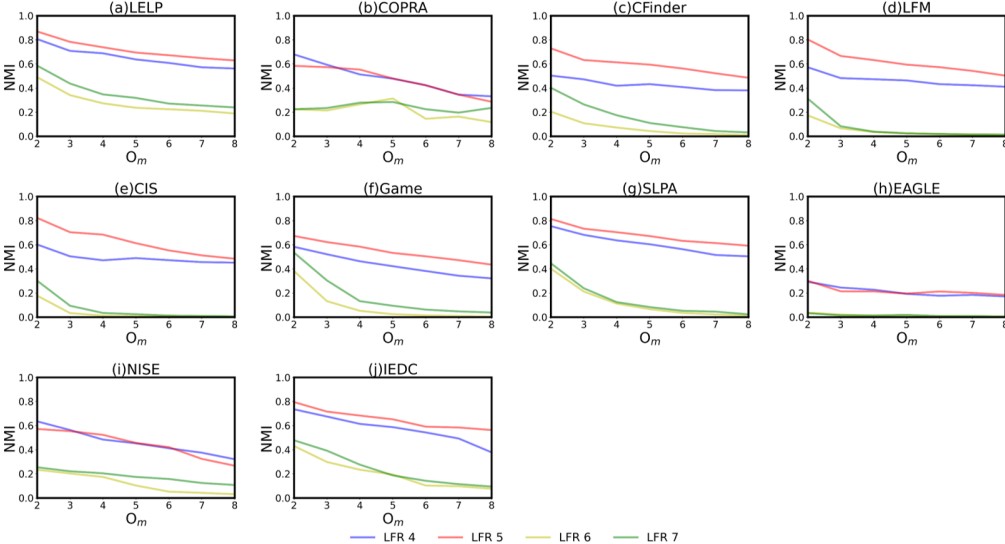

**Figure 3.** Sub figure (**a**–**j**) represent the NMI of different algorithms in the *LFR* 4 to *LFR* 7, where there are different overlapping nodes $O_n$ and community size.

The *NMI* of algorithms on each *LFR* are compared in Figure 4. It can be seen that *LELP* is better in most cases (the blue curve is usually at the top, except Figure 4a,c). However, in the networks with $\mu = 0.1$ the two label propagation algorithms, *COPRA* and *SLPA*, obtain higher *NMI* as a result of the rapid stability of the labels during early propagation. *IDEC* also obtains a better result, which may be because the division of overlapping and non overlapping regions is clear in this case. For the networks where there are many overlapping nodes, *LELP* significantly outperforms other algorithms, excluding *COPRA*. This is because the *closeness* can more accurately reflect the distance between the nodes than the edge, so the impact of complex memberships is not serious. The performance of *Game*, *EAGLE* and *CFinder* are worse than other algorithms in most cases. This shows that algorithms based on cliques are not suitable for large networks where the clique is much smaller than the community.

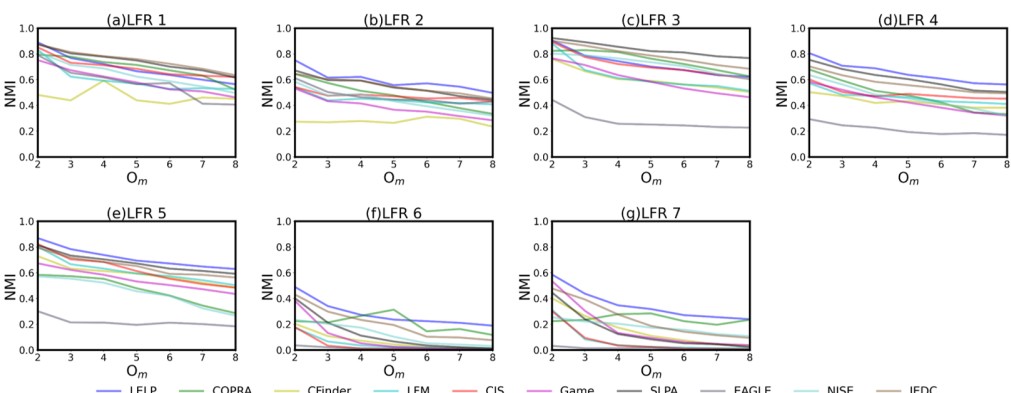

**Figure 4.** Sub figure (**a**–**g**) represent the NMI of different algorithms in the same *LFR*.

*NMI* focus only on providing the measure of overlapping size in each community. It might not be sensitive enough to depict what is going on regarding nodes. It is possible that the overlapping scale detected by algorithms is close to the ground truth, but the overlapping nodes detected are not true. For this reason, measuring the overlapping nodes detected is essential for evaluating the accuracy of detection algorithms. We measure the F-score of algorithms in Figure 5. It can be seen that the F-score of *LELP* is much higher than that of other algorithms when $O_m$ is small. In particular, in *LFR* 4 and *LFR* 5, all algorithms are far behind *LELP*, except for *SLPA* in the case of large $O_m$. In *LFR* 6 and *LFR* 7, *LELP* declines sharply with the increase in $O_m$, but it is still higher than most algorithms. This trend of the F-score is due to the sharp decrease of the *precision* as $O_m$ increases. The reason is that the *precision* of *LELP* is sensitive to $O_m$, which has little influence on *recall*. The *recall* of *LELP* is relatively stable in the face of different $O_m$, and even ascend slightly with the rise in $O_m$. This shows that *LELP* can correctly detect most of the true overlapping nodes, but some non-overlapping nodes are wrongly detected as overlapping nodes. Meanwhile, almost all the non-overlapping nodes are detected correctly.

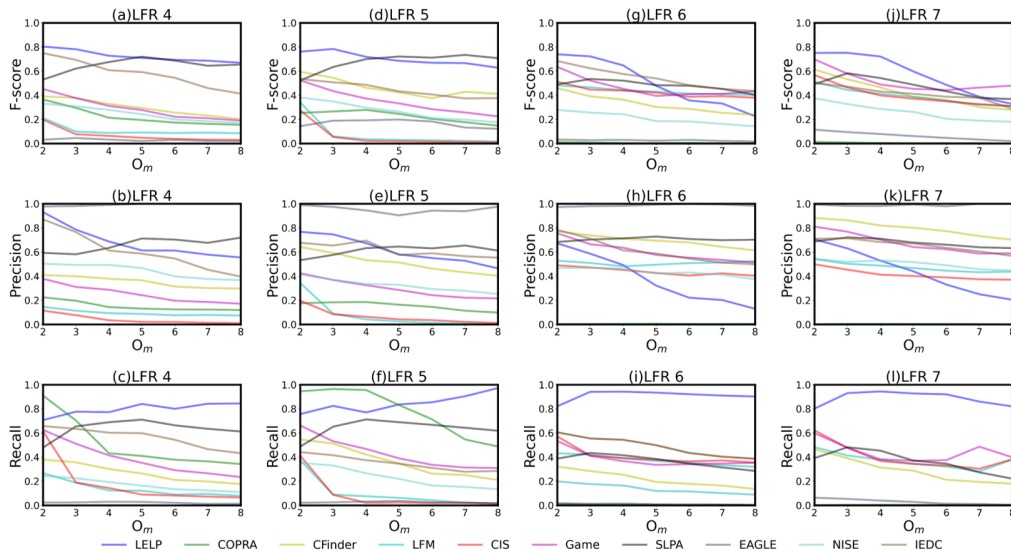

**Figure 5.** Sub figure (**a**,**d**,**g**,**j**) represent the F-score of algorithms in *LFR* 4 to *LFR* 7. Sub figure (**b**,**e**,**h**,**k**) represent the *precision* of algorithms in *LFR* 4 to *LFR* 7. Sub figure (**c**,**f**,**i**,**l**) represent the *recall* of algorithms in *LFR* 4 to *LFR* 7. In *LFR* 4 to *LFR* 7, there are different overlapping nodes $O_n$ and community sizes but other conditions are the same.

### 4.3. Real-World Networks

We tested on a series of real-world networks, whose main information is listed in Table 2. The dolphins and football network are commonly used social network. Email is the core of e-mail communication network from a large European research institution. CA-GrQc is a co-authorship network based on papers in General Relativity publishing in Arxiv. cit-HepTh is the citation graph of high energy physics theory in Arxiv. P2P is the Gnutella peer-to-peer file sharing network. Due to the unknown ground truth of these real-world networks, the algorithms' performance is measured by $Q_{ov}^e$, as shown in Figure 6a. In each sub-figure of Figure 6, networks are arranged from left to right along the *x*-axis, according to the size and symbols of points representing different algorithms. The number of overlapping nodes and the average membership of the nodes are illustrated in Figure 6b,c, respectively.

**Table 2.** The basic features of real-world networks in the experiments are shown in the Table 2, including the number of nodes and edges, average degree, average clustering coefficient and reference.

| Name | $N$ | $E$ | $\bar{k}$ | CC | Ref. |
|---|---|---|---|---|---|
| dolphins | 62 | 3162 | 51 | 0.258 | [33] |
| football | 115 | 1219 | 10.6 | 0.403 | [34] |
| Email | 1005 | 25, 571 | 25.4 | 0.399 | [35] |
| CA-GrQc | 5242 | 14,496 | 2.8 | 0.530 | [36] |
| cit-HepTh | 27,770 | 352,807 | 12.7 | 0.312 | [37] |
| P2P | 62,586 | 147,892 | 2.4 | 0.006 | [36] |

It can be seen that the performance of algorithms in different real-world networks is not constant, but the ranking of algorithms is relatively stable in each network. In most cases, $Q_{ov}^e$ of *LELP* and *SLPA* are generally the best two, especially in large networks, while *EAGLE* and *CIS* are always the worst two. In addition, the performance of *COPRA* is also in the forefront. It illustrates that the algorithms utilizing label propagation are effective because the label propagation is similar to a simulation of information spreading in social networks. Given a good source of propagation, labels are more easily and accurately stable in a community through iterative propagation. In *LELP*, local expansion can generate

some dense regions closer to the center of expected communities as the source of label propagation. This is why our algorithm performs well.

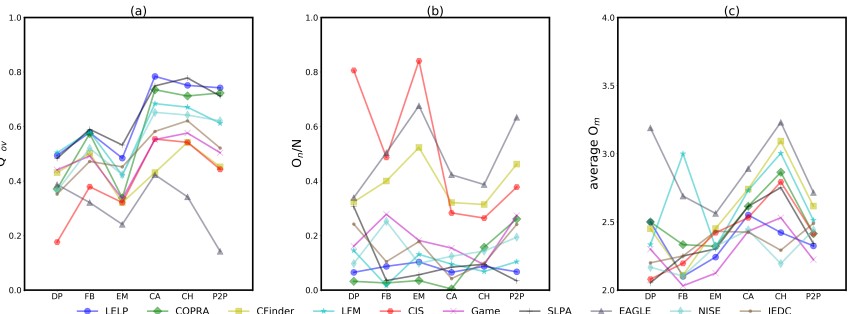

**Figure 6.** Sub figure (**a**–**c**) respectively represent overlapping modularity $Q^e_{ov}$, the proportion of overlapping node $O_n/\mathrm{N}$ and average number of community memberships $Q^m$ of different algorithms in 6 real-world networks.

As can be seen in Figure 6b, some algorithms may over-detect the overlap, such as *CFinder*, *CIS* and *EAGLE*. The proportion of overlapping nodes that they detect in all nodes is far greater than that in other algorithms, and even more than 40% in some cases. This is why these algorithms do not perform well in $Q^e_{ov}$. Too many nodes are detected as overlapping nodes, which indicates that these algorithms fail. In contrast, this proportion of other algorithms is usually below 20%, and even *COPRA* is often lower than 5%. For a detection algorithm, too high of an overlapping proportion detected means that the precision is too rough; too low of a proportion usually means that it does not work. The overlap detected by *LELP* can be considered meaningful because the proportion of overlapping nodes detected in all nodes is appropriate at about 8%. Meanwhile, *LELP*, *LFM*, *NISE* and *SLPA* fluctuate a little as a result of insensitivity to specific structures, and all remain around 10%, except for the small networks. These algorithms also perform well in *LFR*. The stable performance makes it possible for *LELP* to be applied to most networks. Otherwise, the diversity of membership in the tested networks is relatively small, and most nodes have only two or three community memberships, as shown in Figure 6c.

## 5. Conclusions

In this paper, we propose an overlapping community detection algorithm named *LELP* that is a combination of local expansion and label propagation. It first conducts local expansion based on *closeness* to generate some immature communities, which are pruned to retain a dense structure called the dense region. These dense regions are very close to the center of the expected communities and so are taken as the source for label propagation. The experiments in diverse synthetic networks and real-world networks show that *LELP* is excellent, compared with other algorithms. The results in the synthetic networks demonstrate that *LELP* is better than the current mainstream algorithms in general. The performance of *LELP* is less affected by the network size, mixing degree and community size. The proportion of overlapping nodes has a great influence on *LELP*, which is similar to the contrast algorithms. The accuracy of *LELP* is high at the node level. Most of the true overlapping nodes are correctly detected; a few nodes may be misjudged as being overlapping nodes. Almost all non-overlapping nodes are correctly detected. In the real-world networks, *LELP* also performs better than other algorithms. It can be applied to most real-world networks as the result of its stability in the face of different network structures. The number of overlapping nodes detected by *LELP* is in an appropriate range for the detection to be of practical significance.

**Author Contributions:** Conceptualization, X.L. and Q.S.; methodology, X.L. and Q.S.; software, X.L.; validation, X.L.; investigation, X.L. and Q.S.; resources, X.L. and Q.S.; writing—original draft preparation, X.L.; writing—review and editing, X.L. and Q.S.; visualization, X.L.; supervision, Q.S.; project administration, Q.S.; funding acquisition, Q.S. All authors have read and agreed to the published version of the manuscript.

**Funding:** This work was supported in part by the Beijing Social Science Fund No. 11JGB063 and Social Science Research Project of Ministry of Education No. 11YJA630109.

**Data Availability Statement:** Not applicable.

**Conflicts of Interest:** The authors declare no conflict of interest.

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
