# Peer review of "Detect Overlapping Community Based on the Combination of Local Expansion and Label Propagation"

_algorithms, doi:10.3390/a14080237_

Round 1
Reviewer 1 Report
The authors have developed an algorithm to detect overlapping community that is based on two popular algorithms, ie., local expansion and label propagation. The method was applied to both real world as well as synthetic data. For the synthetic data with known community labels, the performance was assessed using F-Score and NMI metrics, while for the real-world data with unknown community labels, the performance was assessed using a different metric. Results were compared with alternate methods.
One of my major concern about the paper is that it is not clear to me how the experimental evaluation is performed. If I am not wrong, to compute F-Score, we measure precision and recall values by splitting the data into test/train/validation set. How this split was performed? Secondly, how many times each experiment was executed. It is important to run the experiment multiple times and report accuracies as well as standard deviation. Finally, can the author explain how some of the hyperparatmeters were adjusted?
Author Response
Thank you for your comments. Here is our reply to your three comments:
(1)reply to "One of my major concern about the paper is that it is not clear to me how the experimental evaluation is performed. If I am not wrong, to compute F-Score, we measure precision and recall values by splitting the data into test/train/validation set. How this split was performed?":
In fact, the ground truth of overlap is known in the experiment of using synthetic network to measure community detection algorithm. In other words, the overlap in an synthetic network is known. Through the community detection algorithm, nodes in this synthetic network will be labeled with community labels and nodes with multiple community labels are considered overlapping. By comparing the ground truth of overlap with that detected by algorithm we can calculate true positive(TP), false positive(FP), false negative(FN) and true negative(TN). Further, we can get precision, recall and f-score. Using F-score to measure the algorithm at the node level is general, such as [1-3].
(2)reply to "Secondly, how many times each experiment was executed. It is important to run the experiment multiple times and report accuracies as well as standard deviation":
We use LFR to generate synthetic networks, and 100 networks are generated for each set of parameters.100 times experiments were carried out in each network and we took the average of the indicators of our algorithm and comparison algorithms. We have stated these in the last paragraph of “4.2. Synthetic Networks”, as “For each set of parameters, we generated 100 instantiations and adopted the average performance of each algorithm.” The statement is not very clear, and we add the following statements, as “For each set of parameters, we generated 100 instantiations and 100 times experiments were carried out on each instantiation. We adopted the average of all these experiments as the algorithm performance.”
(3)reply to "Finally, can the author explain how some of the hyperparatmeters were adjusted?":
The hyperparatmeter involved in our algorithm is only the distance(d) of closeness. During the experiment, we take distance(d) as an integer starting from 1. When distance(d) is an integer greater than 4, the effect of the algorithm changes little. Limited to image space, we only show the performance of the algorithm when distance(d) is 1 to 4. We have stated these in the second paragraph of “4.2. Synthetic Networks”, as “For LELP, we measure its performance with different d, which controls the range of node neighbors in closeness. In general, the local structure information around the nodes is not fully reflected if d is small, while the local details may be ignored if d is so large. Therefore, we set d from 1 to 4 to observe the performance of LELP, as shown in the Figure 1.”
For representative algorithms, the hyperparatmeter that can get the best result is selected. We have stated these in the second paragraph of “4.2. Synthetic Networks”, as “ We compare LELP with some representative algorithms. For algorithms with tunable parameters, the setting that can get the best result is selected.”
Thanks again for your advice. It's very helpful to us.
[1] Xu Z , Liu Y , Jian W , et al. A density based link clustering algorithm for overlapping community detection in networks[J]. Physica A: Statistical Mechanics and its Applications, 2017, 486.
[2] Li Y , He K , Bindel D , et al. Uncovering the Small Community Structure in Large Networks: A Local Spectral Approach[J]. computer science, 2015.
[3] Xie J , Kelley S , Szymanski B K . Overlapping Community Detection in Networks: the State of the Art and Comparative Study[J]. Acm Computing Surveys, 2013, 45(4):1-35.
Reviewer 2 Report
This manuscript mentions about a novel idea of an overlapping community detection algorithm, proposed as LELP, and we run into situations in which individuals experience diverse member relationships in different social clusters.
In this work, authors have appropriately explained the efficacy and feasibility of the algorithms (LELP) proposed therein, by comparing the outperforming feature of the proposed one with others at different levels of LFR and in terms of nodes. The validation seems technically sound with the experiments followed.
The followings are minor/tiny opinions which may be helpful to authors.
1) The authors would enhance the completeness of the manuscript, by checking whether other algorithms such Game, LFM, etc. in Fig.2 as has been further in details introduced and explained either in preceding related works or in Fig.1 through Fig 3.
2) In Table 2, the authors might in details describe the characteristics of all comparison groups such as CA-GrQc, cit-HepTh, P2P, etc.
3) Also, the authors would produce the highly-qualified publications, by conducting extra tiny works of proof-reading and text editing.
Thank you.
Author Response
Thank you for your comments. Here is our reply to your three comments:
(1)reply to “The authors would enhance the completeness of the manuscript, by checking whether other algorithms such Game, LFM, etc. in Fig.2 as has been further in details introduced and explained either in preceding related works or in Fig.1 through Fig 3.”:
Although we use many comparison algorithms, we have classified them into four categories in Chapter 2 according to characteristics.
In the local expansion algorithm, the form of fitness function used by most algorithms is similar. We introduced CIS and LFM in relative detail. MONC and NISE select other variables based on the form of fitness functions. Therefore, considering the length of the article, we did not use the formula to show these differences, but described it in words. Similarly, COPRA and SLPA are the same condition in the label propagation algorithm. We focus on these two kinds of algorithms, because part of our algorithm comes from their ideas.
We introduce the Clique percolation algorithm and other algorithms in less space. Clique percolation method, such as CPM and EAGLE, are based on the idea of aggregation to detect communities, and pay more attention to the aggregation rules. The algorithm is relatively refined and simple, so We use less space to describe them. Other algorithms mentioned in this paper have various forms. We only introduced their core ideas and key steps.
(2)reply to “In Table 2, the authors might in details describe the characteristics of all comparison groups such as CA-GrQc, cit-HepTh, P2P, etc.”:
We supplement some details of the network in the first paragraph of 4.3. Real-world Networks, as “The dolphins and football network are commonly used social network. Email is the core of E-mail communication network from a large European research institution. CA-GrQc is a co-authorship network based on papers in General Relativity publishing in Arxiv. cit-HepTh is the citation graph of high energy physics theory in Arxiv. P2P is the Gnutella peer-to-peer file sharing network.”
(3)reply to “Also, the authors would produce the highly-qualified publications, by conducting extra tiny works of proof-reading and text editing.”:
We checked the manuscript again and modified some grammar and vocabulary to expect our expression to be more accurate. The modified part has been highlighted in the revision.
Thanks again for your advice. It's very helpful to us.